

# Microbial methane formation in deep aquifers associated with the sediment burial history at a coastal site

Taiki Katayama[1], Reo Ikawa[2], Masaru Koshigai[2], Susumu Sakata[1]

[1]Geomicrobiology Research Group, Institute for Geo-Resources and Environment, Geological Survey of Japan (GSJ), National Institute of Advanced Industrial Science and Technology (AIST), 1-1-1 Higashi, Tsukuba, 305-8567, Japan.
[2]Groundwater Research Group, Institute for Geo-Resources and Environment, GSJ, AIST, 1-1-1 Higashi, Tsukuba, 305-8567, Japan.

*Correspondence to*: Taiki Katayama (katayama.t@aist.go.jp) and Susumu Sakata (su-sakata@aist.go.jp)

**Abstract.** Elucidating the mechanisms underlying microbial methane formation in subsurface environments is essential to
understand the global carbon cycle and to explore natural gas deposits. This study examined how microbial methane formation
(i.e. methanogenesis) occurs in natural gas-bearing sedimentary aquifers throughout the sediment burial history. Water samples
collected from six aquifers of different depths exhibited ascending vertical gradients in salinity from brine to freshwater and
in temperature from mesophilic to psychrophilic conditions. Analyses of gas and water isotopic ratios and microbial
communities indicated the predominance of methanogenesis via $CO_2$ reduction. However, the hydrogen isotopic ratio of water
changed along the depth and salinity gradient, whereas the ratio of methane changed little, suggesting that *in situ*
methanogenesis in shallow sediments does not significantly contribute to the methane in the aquifers. The population of
methane-producing microorganisms (methanogens) was highest in the deepest saline aquifers, where the water temperature,
salinity, and the total organic carbon content of the adjacent mud sediments were highest. Cultivation of the hydrogenotrophic
methanogens that dominated in the aquifers showed that the methanogenesis rate was maximized at the temperature
corresponding to that of the deepest aquifer. These results suggest that high-temperature conditions in deeply buried sediments
are associated with enhanced in situ methanogenesis, and that methane formed in the deepest aquifer migrates upwards into
the shallower aquifers by diffusion.

## 1 Introduction

Terrestrial subsurface environments are massive reservoirs of water and organic matter, and they harbor a large fraction of the
microorganisms present on Earth (McMahon and Parnell, 2014; Magnabosco et al., 2018). Aquifers formed in sedimentary
environments provide microorganisms with pore spaces, water, and the buried organic materials that serve as energy and
carbon sources, thereby sustaining metabolic activity and influencing the organic and inorganic geochemistry of subsurface
environments (McMahon and Chapelle, 1991; Lovley and Chapelle, 1995; Fredrickson et al., 1997; Krumholz et al., 1997).

Methanogenesis, biological methane formation, is a terminal process involving the degradation of organic matter in anoxic
environments where electron acceptors other than $CO_2$ are depleted. Methanogens comprise a diverse group of archaea that
produce methane from $H_2$ and $CO_2$ (hydrogenotrophic), methylated compounds (methylotrophic), or acetate (acetoclastic).
Because active methanogens are widespread in subsurface environments (Mesle et al., 2013), it has been speculated that



microbial methane may comprise a larger proportion of natural gas reserves than previously thought (Kotelnikova, 2002).

Microbial methane has been estimated to account for more than 20% of global natural gas resources (Katz, 2011).

The sedimentary aquifers explored in this study are located beneath the Teshio Plain, in a coastal area of northern Japan (Fig. S1). Isotopic analysis of hydrocarbon gases in this area has revealed that methane predominates over ethane and propane, thus suggesting a microbial origin for the natural gases (Tamamura et al., 2014). Ikawa et al. (2014), who conducted geochemical analyses of porewaters extracted from sediment core samples from the D-1 borehole, drilled to a depth of 1,000

m below the ground surface (mbgs) of the plain, found vertical gradients of the Cl$^-$ concentration and the hydrogen isotopic ratio of water (Fig. 1b, c). They proposed the following processes to explain how these gradients formed. Sediments corresponding to the Yuchi Formation were deposited in shallow-marine environments during the late Pliocene Epoch. Deposition of the sediments corresponding to the Sarabetsu Formation, which overlies the Yuchi Formation, occurred in bay, lagoon, or fluvial environments, during the early Pleistocene Epoch. Brackish and fresh waters trapped during this period

became mixed with brine from the Yuchi Formation by diffusion, resulting in the formation of a continuous vertical salinity gradient. Aquifers within the upper part of the Sarabetsu Formation (90–280 mbgs) were recharged with paleo-meteoric water. Throughout the burial history of the Yuchi Formation, water salinity decreased while temperature increased along the geothermal gradient (Fig. 1d). Therefore, aquifers in the Yuchi Formation provide an opportunity to explore the impacts of these geochemical changes on microbial methane formation.

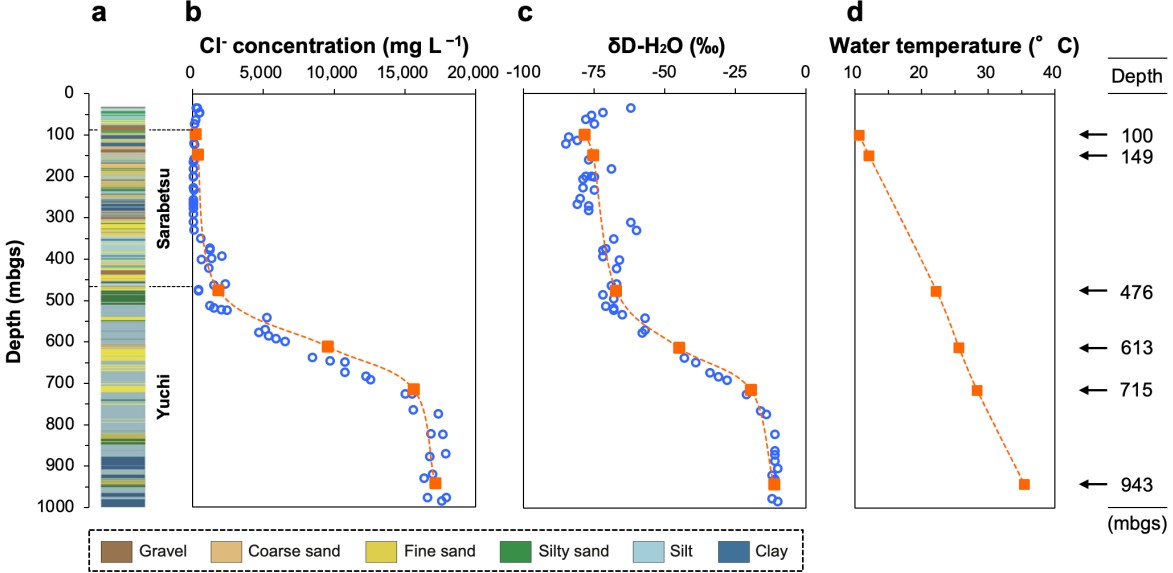


**Figure 1.** Depth profiles of borehole D-1: (**a**) lithology, (**b**) Cl$^-$ concentration, (**c**) hydrogen isotopic ratio, and (**d**) temperature of sediment core porewater (blue open circles) (data from Ikawa et al., 2014) and water samples from the aquifers investigated in this study (orange solid squares). The aquifer depths at which the waters were sampled are shown at the right.



In this study, water samples were collected from the saline aquifers in the Yuchi Formation (at 476, 613, 715, and 943

mbgs) and from the freshwater aquifers in the upper part of the Sarabetsu Formation (at 100 and 149 mbgs), as reference

material for the Yuchi Formation samples (Fig. 1). Whereas many previous studies have conducted geologic and geochemical

analyses to examine the relationship between the burial history of geological formations and the occurrence of methane

deposits [e.g. Zhang et al. (2013) and references therein], this study combines microbiological analyses (e.g. gene-sequencing

and cultivation-based analyses) with geochemical analyses to elucidate this relationship from multiple perspectives.


## 2 Materials and methods

### 2.1 Site description and sample collection

The study site was located on a sand dune 300 m from the coastline at an elevation of 5.2 m above sea level (44.9948° N,

141.6882° E) (Fig. S1). The geology of this site down to the depth of 1,000 mbgs consists of the Yuchi Formation, the Sarabetsu

Formation, and alluvium, in ascending order. Hydraulic gradient and conductivity, water isotopic, and geological data indicate

that clay aquitards at approximately 300 mbgs prevent water in the upper part of the Sarabetsu Formation from mixing with

connate water in the lower part of the Sarabetsu Formation and the underlying Yuchi Formation (Ikawa et al., 2014). Indeed,

a drastic change observed in the stable hydrogen ($\delta$D) isotope ratio of the water at approximately 300 mbgs (Fig. 1c) indicates

that the aquifers above and below that depth are hydrologically different (Ikawa et al., 2014).

The water samples for this study were collected from three wells, D-1, D-2, and D-3. The D-2 and D-3 wells are located

within 30 m of D-1. Two samples from D-2 and D-3 are freshwaters derived from the upper Sarabetsu Formation, and four

samples from D-1 are brines derived from the Yuchi Formation. Samples from D-2 and D-3 were pumped through strainers at

the depths of 90-100 mbgs and 130-149 mbgs, respectively, while samples from D-1 were pumped with borehole packer

assemblies at the depths of 476, 613, 715 and 943 mbgs. The water samples were obtained after chemical parameters, such as

water temperature, electrical conductivity, pH, and oxidation-reduction potential, had stabilized. Before the sample collection,

the waters of approximately 44–160 times the wellbore volume of D-2 or D-3 and 4–13 times the packer-sealed area of D-1

were pumped out.

Samples for microbial cultures were collected in sterilized glass bottles with butyl rubber stoppers and screw caps. The

bottles were purged with $N_2$ gas before and during the sample collection and then filled with water to maintain the samples

under anaerobic conditions. For the molecular analysis, 4-L water samples were collected and filtered through a 0.2-$\mu$m-pore-

size Millipore Express Plus membrane filter (Millipore, Billerica, MA, USA) and stored at −20 °C. The samples used for total

cell counts were fixed with formalin at a final concentration of 2% (v/v) immediately after sampling and stored at 4 °C. The

gases that were associated with the water and naturally separated under atmospheric pressure at the time of sampling were

collected over water.


### 2.2 Geochemical analysis





The chemical compositions and stable hydrogen isotope ratios (δD) of the water samples were measured by using ion chromatography (DIONEX ICS-5000, Thermo Fisher Scientific, Bremen, Germany) and a liquid water isotope analyzer (L2120-i, Picarro, Santa Clara, CA, USA), respectively. The standard deviation of δD for water was 1‰.

The gas composition was measured by using a gas chromatograph (GC) with a flame-ionization detector and thermal conductivity detector (TCD). The stable carbon ($\delta^{13}C$) and hydrogen (δD) isotope ratios of methane and $\delta^{13}C$ of carbon dioxide were measured with a Trace Ultra gas chromatograph connected to a DELTA V plus isotope ratio mass spectrometer (IRMS) via a GC IsoLink combustion/pyrolysis interface (Thermo Fisher Scientific). The Natural Gas Standard NGS3 was used as an isotope reference material. The standard deviations of δD and $\delta^{13}C$ for methane were 1.6‰ and 0.3‰, respectively, and the

standard deviation of $\delta^{13}C$ for carbon dioxide was 0.2‰.

The total organic carbon (TOC) content of silty or clayey sediment core samples from the Yuchi Formation, collected previously by Ikawa et al. (2014) was measured with a TruSpec CHN analyzer (LECO). Before the measurements, samples were pulverized to less than 200 mesh and treated with 1M HCl to remove inorganic carbon.

## 2.3 Direct cell counts

A fixed water sample was filtered through a 0.2-μm-pore-size Isopore membrane filter (Millipore), stained for 10 min with SYBR Green solution (10 μg mL$^{-1}$), and observed under an epifluorescence microscope (Olympus, Tokyo, Japan).

## 2.4 DNA extraction and quantitative PCR for 16S rRNA and *mcrA* genes

DNA was extracted from the filtered water and methanogenic culture (as described below) samples by using a PowerWater kit (MoBio Laboratories, CA, USA) according to the manufacturer's protocol. Quantitative PCR targeting bacterial and archaeal 16S rRNA genes in water samples was performed in triplicate by the quenching probe method (Tani et al., 2009) using TITANIUM Taq DNA polymerase (Takara, Otsu, Japan) in a Rotor-Gene Q real-time cycler (QIAGEN, Valencia, CA). The primers and probes used for real-time PCR and sequencing (as described below) are listed in Table S1. The cycling

conditions were 95 °C for 2 min, followed by 50 cycles of 93 °C for 15 s, 61 °C for 20 s, and 72 °C for 25 s. The copy numbers of the *mcrA* gene, which encodes a methyl-coenzyme M reductase alpha subunit, a enzyme central to the methanogenesis, were quantified in triplicate by SYBR Green real-time PCR using a SYBR Premix Ex-Taq II (Takara) in a LightCycler 1.0 (Roche, Basel, Switzerland). The cycling conditions were 95 °C for 30 s, followed by 50 cycles of 95 °C for 15 s, 52 °C for 20 s, and 72 °C for 25 s. Ten-fold serial dilutions of the target PCR products for *Escherichia coli* K12 (ATCC 10798) (for the

bacterial 16S rRNA gene) and *Methanobacterium bryantii* M.o.H. (ATCC 33272) (for the archaeal 16S rRNA and *mcrA* genes) were also amplified to calculate the gene copy numbers.

## 2.5 454 pyrosequencing of 16S rRNA genes

The 16S rRNA genes, including the V3 and V4 regions, were amplified using AmpliTaq Gold 360 DNA polymerase (Life

Technologies, CA, USA) with a Univ515F primer (fused to the 454-specific adaptor A and 6-nt barcode sequences) and a



Univ926R primer (fused to adaptor B). Cycling conditions were 95 °C for 10 min, followed by 25–27 cycles of 95 °C for 30 s, 50 °C for 40 s, and 72 °C for 30 s, and a final extension period of 7 min at 72 °C. Four replicates of PCR products for each sample were pooled and purified by using the MonoFas DNA purification kit. Pyrosequencing was performed using a 454 Life Sciences GS FLX Titanium platform (Roche, Basel, Switzerland) at Hokkaido System Science Co., Ltd. (Sapporo, Japan).


## 2.6 Cloning and Sanger sequencing of the *mcrA* gene

The *mcrA* gene was amplified from the six original water and methanogenic culture samples (as described below) by using the MLf and MLr primer pair (Luton et al., 2002) and AmpliTaq Gold 360 DNA Polymerase (ThermoFisher Scientific). The PCR products were purified by using a MonoFas DNA purification kit (GL Sciences, Tokyo, Japan), cloned in the pCR4-TOPO

vector (ThermoFisher Scientific), and sequenced by the dideoxynucleotide chain-termination method using BigDye terminator reagents (ThermoFisher Scientific) and an automated sequence analyzer (3730 DNA Analyzer, ThermoFisher Scientific) according to the manufacturer's instructions.

## 2.7 Sequence analysis

The 454 pyrosequencing reads of the 16S rRNA genes were analyzed by using Mothur ver. 1.48 software (Schloss et al., 2009) as described previously (Katayama et al., 2015, 2022) with the following modifications. Quality-filtered sequences with an average length of 250 bp were classified by using a Bayesian classifier based on the Silva taxonomy SSU Ref 138.1 dataset (Pruesse et al., 2007) with a confidence threshold of 80%. The putative methanogens in the 16S rRNA gene sequences were searched based on this taxonomic classification.

Sanger sequences of the *mcrA* gene were translated to amino acid in silico and aligned by using MAFFT ver. 7 software (Katoh and Standley, 2013). Amino acid sequences with >93% sequence identity were treated as operational taxonomic units (OTUs). In each OTU, the most abundant sequence was selected as the representative sequence. The most closely related species to the OTUs were searched by using BLAST (http://blast.ncbi.nlm.nih.gov/Blast.cgi).

The 454-sequencing data were submitted to the DDBJ Sequence Read Archive database under accession number

DRA001113. The GenBank/EMBL/DDBJ accession numbers for the *mcrA* gene sequences are LC214911 to LC214935.

## 2.8 Cultivation of methanogens

The basal medium used for the methanogenic cultures consisted of 10 mM $NH_4Cl$, 1 mM $KH_2PO_4$, 15 mM $MgCl_2·6H_2O$, 1 mM $CaCl_2·2H_2O$, 30 mM $NaHCO_3$, 1 mL $L^{-1}$ of selenium and tungsten solution, 1 mL $L^{-1}$ of trace elements solution, 2 mL

$L^{-1}$ of vitamin solution, 1 mL $L^{-1}$ of resazurin solution (1 mg $mL^{-1}$), and 0.5 mM titanium(III) nitrilotriacetate (as a reducing agent) (Katayama and Kamagata, 2018; Katayama et al., 2020). Twenty milliliters of basal mineral medium was dispensed into 67-mL serum vials. The vials were sealed with butyl rubber septa and aluminum crimps under an atmosphere of $N_2/CO_2$ (80:20, v/v). The medium was supplemented with either $H_2/CO_2$ (80:20, v/v; 0.1 MPa) or acetate (20 mM) as methanogenic substrates. The medium was further supplemented with NaCl at final concentrations of 250 and 500 mM, which approximated



its in situ concentrations at 613 and 943 mbgs, respectively. One-milliliter aliquots of the water samples from 613 and 943 mbgs were dispensed into each medium and incubated at 25 °C (for the 613-mbgs sample) and 35 °C (for the 943-mbgs sample) to approximate the in situ water temperature. Methane production was measured using a GC equipped with a TCD. After methane production was terminated, 4 mL of the sample cultures were harvested by filtration and the *mcrA* gene was cloned and sequenced as described above.


**2.9 Effects of salinity and temperature on methanogenesis**

Methane-producing cultures supplemented with $H_2/CO_2$ or acetate from the 943-mbgs water sample were subsequently inoculated into fresh medium to examine the methanogenic activity under different salinity and temperature conditions. Cultures with different salinities were grown in a basal mineral medium containing 15, 270, or 480 mM $Cl^-$ at 35 °C. Cultures

with different temperatures were grown in basal mineral medium containing 480 mM $Cl^-$ at 20, 25, 35, or 45 °C. Both culture sets were supplemented with $H_2/CO_2$ (80:20, v/v; 0.1 MPa) or acetate (20 mM). The time course for methane production was determined to calculate the methane production rate.

**2.10 Cultivation of microorganisms syntrophically oxidizing acetate to methane**

Semi-continuous cultivation supplemented with a low concentration of acetate (0.4 mM) was performed in a modified 132-mL glass vial containing sterilized pieces of non-woven fabric as the carrier material for microbial cells (Fig. S2a) to culture microorganisms involved in syntrophic acetate oxidation (SAO) coupled to methanogenesis via carbonate reduction. Forty milliliters of basal mineral medium (as described above) supplemented with NaCl (500 mM) and acetate (0.4 mM) was dispensed into the vial. Ten milliliters of the 943-mbgs water sample was used as an inoculum. The top and bottom of the vial

were sealed with a butyl rubber septum and aluminum crimps, and the culture was incubated at 35 °C under an $N_2/CO_2$ (80:20, v/v) atmosphere for 10 months. During cultivation, the culture was manually fed with acetate (0.4 mM) at 3-week intervals. Before feeding, 20 mL of culture liquid was removed from the vial through the bottom septum using a needle syringe, and 20 mL of fresh medium containing acetate (final concentration: 0.4 mM) was then added to the vial through the top septum so that the syntrophic association of microbial cells was not physically disrupted by turning the vial upside down.

After cultivation, 2 mL of culture liquid and a piece of non-woven fabric were transferred from the semi-continuous cultivation system to 67-mL serum vials containing basal mineral medium with 0.4 mM [2-$^{13}$C]-acetate or non-labeled acetate (used as a control) to determine the presence of SAO activity. In both cultures, 0.4 mM labeled or unlabeled acetate supplement was added at 2-week intervals. The incubation was performed in duplicate. Time courses for methane production and the stable carbon isotopic ratio of dissolved inorganic carbon (DIC) in the culture liquid were determined by using a GC and a GC/IRMS,

respectively, as described above.

**3 Results**

**3.1 Geochemistry of water and sediment**



The geochemical properties of the six water samples are summarized in Tables 1 and 2. The redox potentials in the freshwater
samples from the upper Sarabetsu Formation aquifers (100 and 149 mbgs) were higher (> −210 mV) than those in the brine
samples from the Yuchi Formation aquifers (476, 613, 715, and 943 mbgs) (< −290 mV). $NO_3^-$ and $SO_4^{2-}$ were detected only
in the upper Sarabetsu Formation samples, but mostly in small amounts not exceeding 1.4 mg $L^{-1}$. The water temperature
increased with depth ($r > 0.99$, $p < 0.001$; linear regression $t$-test) and ranged from 10.6 to 35.4 °C. Stiff diagrams show a
difference in water chemistry between the upper Sarabetsu and the Yuchi Formation samples (Fig. S3), which is consistent
with hydrologic separation above and below the clay aquitards (Ikawa et al., 2014). The $Cl^-$ concentrations and $\delta D$-$H_2O$ values
of all six water samples were similar to those of porewater within the sediment cores at the corresponding depths (Fig. 1); thus,
they show no sign of cross-contamination among the samples.

In all water samples, $CH_4$ accounted for approximately >75% of the total dissolved gas (Table 2). The proportions of $CH_4$
and $CO_2$ increased with depth, whereas that of $N_2$ decreased. The stable carbon ($\delta^{13}C$) and hydrogen ($\delta D$) isotopic ratios of
methane ranged from −77.1‰ to −67.5‰ and from −258‰ to −196‰, respectively. Among the samples from the Yuchi
Formation, changes in $\delta D$-$CH_4$ values were small (7‰) compared with changes in $\delta D$-$H_2O$ values (56‰) (also evident in Fig.
2c). Plots of the isotopic ratios of the gases and water, $\delta^{13}C$-$CH_4$ versus $\delta D$-$CH_4$, $\delta^{13}C$-$CO_2$ versus $\delta^{13}C$-$CH_4$, and $\delta D$-$CH_4$
versus $\delta D$-$H_2O$ (Whiticar, 1999) (Fig. 2a–c), indicated a microbial origin of dissolved methane via the $CO_2$ reduction pathway
in both the upper Sarabetsu and the Yuchi Formation samples. Methane dissolved in water from the Koetoi Formation, which
underlies the Yuchi Formation (Fig. S1), plotted near the boundary between a biogenic and a thermogenic origin (Tamamura
et al., 2014) (Fig. 2a). The lack of thermogenic methane produced at great depth in the upper Sarabetsu and Yuchi Formation
samples implies that methanogenesis occurred within these formations.





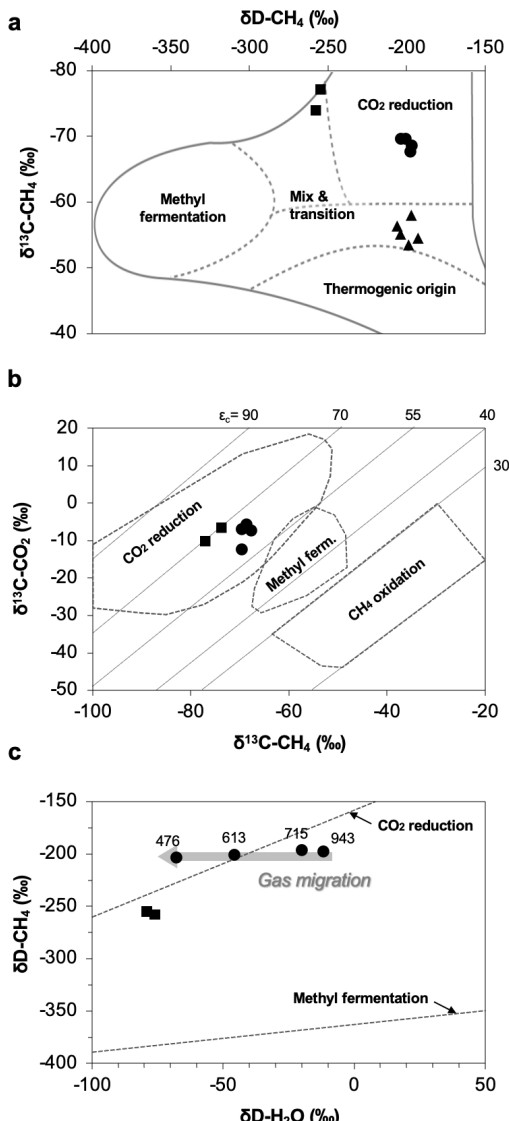

**Figure 2.** Relationships between (**a**) δD and δ¹³C of methane, (**b**) δ¹³C of methane and δ¹³C of carbon dioxide, and (**c**) δD of methane and δD of water for water samples from the upper Sarabetsu Formation (squares) and the Yuchi Formation (circles). The origins of methane are estimated in each plot according to Whiticar (1999). In (**a**), data for water samples from the Koetoi Formation (triangles) (Tamamura et al. 2014) are shown for comparison. The light gray diagonal lines in (**b**) indicate carbon isotopic fractionation contours ($\varepsilon_c \approx \delta^{13}C\text{-}CO_2 - \delta^{13}C\text{-}CH_4$). In (**c**), the depths (mbgs) of water samples from the Yuchi Formation are indicated.



The TOC content in sediment core samples from the Yuchi Formation ranges from less than 0.1% to more than 0.5% (Fig. S4). Despite some dispersion associated with lithological changes, TOC shows an overall increasing trend with increasing depth ($r = 0.77$, $p < 0.001$).

**3.2 Enumeration of total microbial cells and of 16S rRNA and *mcrA* genes**

The number of microbial cells in the water samples ranged from $1.8 \times 10^4$ to $1.5 \times 10^6$ cells mL$^{-1}$ (Fig. 3a). The highest and lowest numbers were measured in the 100- and 715-mbgs samples, respectively, and the microbial cell densities in the studied aquifers are comparable to those reported in other deep aquifers ($10^2$ to $10^6$ cells mL$^{-1}$) (Pedersen, 1993).

Bacterial and archaeal populations were measured by quantitative real-time PCR (Fig. 3a). Copy numbers of the bacterial 16S rRNA gene were $10^3$–$10^5$ mL$^{-1}$, whereas those of archaea were $10^2$–$10^4$ mL$^{-1}$. In the Yuchi Formation, copy numbers of bacterial and archaeal genes were highest in the deepest (943-mbgs) sample. The copy number of the *mcrA* gene, which is used to estimate methanogen populations, was also highest in this sample at $3.5 \times 10^3$ gene copies mL$^{-1}$, which is 2–3 orders of magnitude higher than the number of copies of the *mcrA* gene in the other samples.

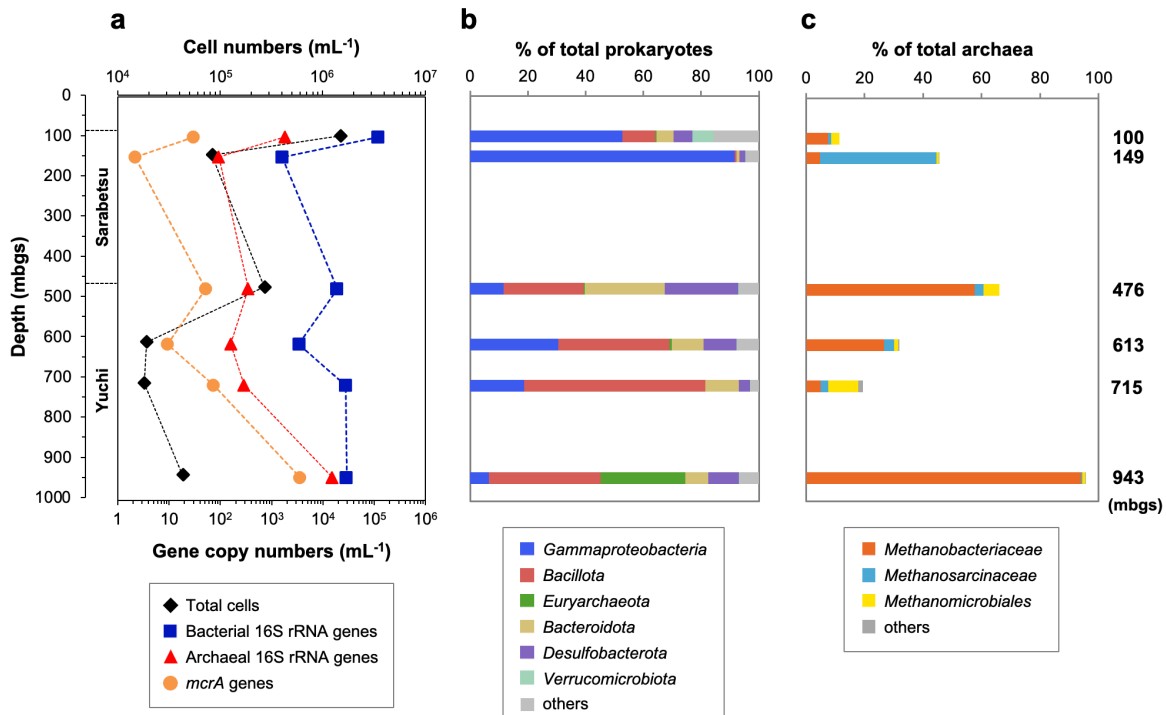

**Figure 3.** Depth-related changes in (**a**) microbial populations and (**b**) prokaryotic and (**c**) methanogenic community compositions, based on the 16S rRNA gene sequences in water samples.

**3.3 Microbial community compositions in the water samples**



The 454-pyrosequencing analysis of 16S rRNA genes was performed to examine the compositions of prokaryotic and methanogenic communities and their consistency with the *mcrA* gene sequencing analysis results, as described below. After quality filtering, the pyrosequencing reads yielded 14,304–43,532 reads per sample. The major taxonomic groups (>5% of the total reads in at least one sample) belonged to the following phyla or classes: *Gammaproteobacteria*, *Bacillota* (formerly *Firmicutes*), *Euryarchaeota*, *Bacteroidota*, *Desulfobacterota*, and *Verrucomicrobiota* (Fig. 3b). *Gammaproteobacteria* sequences were more abundant in the upper Sarabetsu Formation samples, whereas *Bacillota* and *Bacteroidota* were more abundant in the Yuchi Formation samples.

The sequences assigned to putative methanogens accounted for 0.2%–30% and 11%–95% of prokaryotic and archaeal 16S RNA gene sequences (Fig. 3c), respectively. The high proportion of methanogenic sequences in the 943-mbgs sample is consistent with the quantitative PCR results. Sequences assigned to hydrogenotrophic *Methanobacteriales* were commonly detected at all depths. In the 149-mbgs sample, high proportions of acetoclastic and/or methylotrophic methanogens of the genus *Methanosarcina* (*Methanosarcinales*) were detected.

### 3.4 Methanogen diversity in the water samples based on the *mcrA* genes

A total of 64–69 clones of the *mcrA* gene per sample were grouped into 14 OTUs (Table 3). Similar to the 16S rRNA gene sequencing analysis results, methanogen diversity differed between the upper Sarabetsu and Yuchi Formation samples. Sequences related to acetoclastic *Methanosaeta* and hydrogenotrophic *Methanoregula* were abundant in the upper Sarabetsu Formation samples, whereas hydrogenotrophic *Methanobacterium* sequences were abundant in the Yuchi Formation samples. In the upper Sarabetsu Formation samples, *Candidatus* Methanoperedens, which oxidizes methane by coupling to nitrate reduction (Haroon et al., 2013), was also detected. Despite its high proportion, a shift in carbon isotope values towards the methane oxidation region on the $\delta^{13}$C-CH$_4$ versus $\delta^{13}$C-CO$_2$ plot (Whiticar, 1999) was not observed in those samples (Fig. 2b).

### 3.5 Methanogen diversity in cultures

The 613- and 943-mbgs water samples from the Yuchi Formation aquifers were cultured with methanogenic substrates (i.e. H$_2$/CO$_2$ or acetate) under the in situ salinity and temperature conditions to obtain culturable methanogens. Methane was produced from all samples. More than 82% (v/v) of the maximum theoretical yield of methane was obtained, indicating that the supplied methanogenic substrates were primarily used for methanogenesis.

In the acetate-amended cultures, *Methanosarcina* dominated, whereas *Methanoculleus* and *Methanobacterium* were detected in large proportions in the H$_2$/CO$_2$-amended cultures (Table 3). The sequences of these taxa were almost identical to those obtained directly from the original water samples, indicating that the predominant methanogens inhabiting the saline aquifers were successfully cultured. The diversity of the culturable methanogens was not clearly different between the 613- and 943-mbgs samples.

### 3.6 Effects of salinity and temperature on methanogenic activity





The methanogen cultures from the 943-mbgs water sample were subsequently cultured under different salinity and temperature conditions based on their depth profiles in the Yuchi Formation (Fig. 1b, d). In the $H_2/CO_2$-supplemented cultures, the methane production rate decreased slightly under the lowest salinity condition (i.e. 15 mM $Cl^-$), whereas a more notable decrease was observed under the highest salinity condition (480 mM $Cl^-$) in the acetate-supplemented cultures (Fig. 4a). Temperature changes more drastically affected methanogenic activity than salinity changes (Fig. 4b). In both $H_2/CO_2$- and acetate-supplemented cultures, methane producation rates were highest at 35 °C. At a temperature of 45 °C, which approximately corresponds to that at the depth of 1275 mbgs (assuming a thermal gradient of 2.92 °C per 100 m; Fig. 1d), methane production was observed only in the $H_2/CO_2$-amended culture.

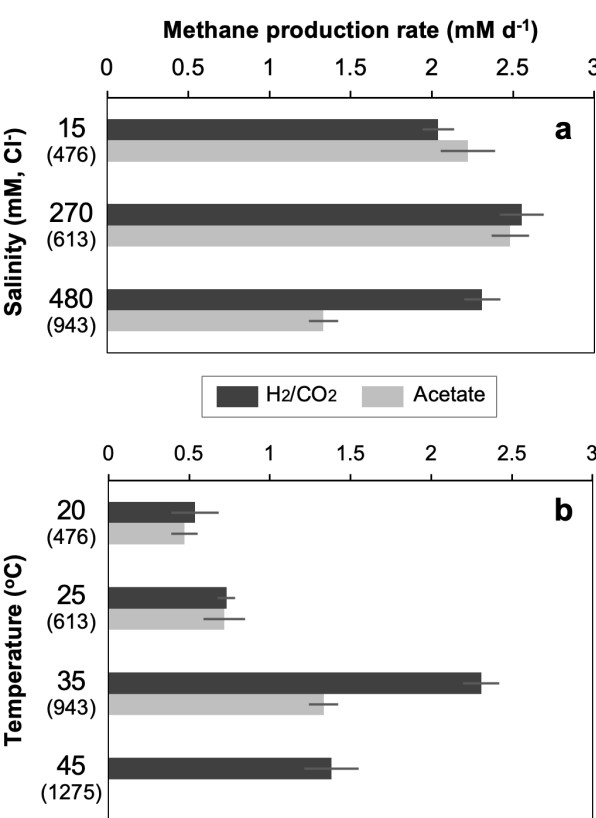

**Figure 4.** Effects of depth-related salinity (**a**) and temperature (**b**) changes on methane production rates in $H_2/CO_2$- and acetate-amended microcosms in the 943-mbgs water sample. The cultivation temperature for (**a**) was 35 °C, and the $Cl^-$ concentration in the culture medium for (**b**) was 480 mM. The values in parentheses are the depths (mbgs) of the saline aquifers corresponding to each culture condition.

**3.7 The potential for syntrophic acetate oxidation (SAO) coupled with hydrogenotrophic methanogenesis**



The SAO activity coupled with hydrogenotrophic methanogenesis (Zinder and Koch, 1984) in the 943-mbgs water sample was assessed by semi-continuous cultivation of SAO microorganisms fed with a low concentration of acetate (Shigematsu et al.,

2004) (Fig. S2a). After 10 months of cultivation, SAO activity was measured by using [2-$^{13}$C]-acetate.

Methane was produced stoichiometrically from acetate in cultures supplemented with labeled and non-labeled acetate (Fig. S2b). Values of $\delta^{13}$C-DIC increased from approximately −25‰ to 9‰ over time in the culture supplemented with [2-$^{13}$C]-acetate, whereas no significant change was observed in the culture with non-labeled acetate (Fig. S2c), clearly indicating SAO activity: in the acetoclastic methanogenic pathway, the methyl group of acetate is converted to methane but not to $CO_2$ (Ferry,

1993), whereas the methyl group of acetate is converted to $CO_2$ and subsequently to methane when SAO is coupled with hydrogenotrophic methanogenesis (Zinder and Koch, 1984).

## 4 Discussion

This study examined microbial methane formation in relation to geochemical changes in deep sedimentary environments. Gas

isotope analysis results suggested that methanogenesis occurred mostly via a carbonate reduction pathway in the Yuchi Formation. This finding is consistent with sequencing analysis results showing the predominance of hydrogenotrophic methanogens. In this formation, the isotopic ratio of hydrogen in water changed with depth and was coupled with a decrease in salinity due to diffusive mixing of brine with freshwater from the overlying formation (Ikawa et al., 2014). If substantial methanogenesis had occurred via the $CO_2$ reduction pathway after this dilution, the $\delta$D-$CH_4$ value would have changed along

with the $\delta$D-$H_2$O value and become distinct from the value of the deepest 943-mbgs sample, that is, the least diluted brine. This change would have occurred because all hydrogen atoms in methane produced via the $CO_2$ reduction pathway are derived from the ambient water (Daniels et al., 1980). However, the results showed almost no change in $\delta$D-$CH_4$ compared with $\delta$D-$H_2$O, suggesting that *in situ* methanogenesis in the shallow part of this formation does not contribute significantly to methane deposits overall and that methane produced in the deeper layers of the Yuchi Formation migrated upward in association with

the diffusive mixing of brine with freshwater (Fig. 2c).

This interpretation is supported by the experimental results. A remarkably high methanogen population, composed primarily of hydrogenotrophic methanogens, was observed in the deepest brine sample. In addition, hydrogenotrophic methanogensis was estimated to proceed faster in deeper aquifers in the Yuchi Formation, where salinity and temperature are higher. The TOC content of the sediment core samples from this formation increased with depth, and the sediments adjacent

to the deepest aquifer contained 2 to 3 times as much TOC as those adjacent to other, shallower aquifers (Fig. S4). Similar to our results, the population of microorganisms, including methanogens represented by their lipid biomarkers, locally increases with increasing TOC content in deep marine sediments (Cragg et al., 1996; Oba et al., 2015). Previous studies indicated that low-permeability sediments are rich in organic materials and their fermentation products, such as acetate, diffuse into adjacent, more permeable aquifers, where they are consumed by microorganisms (McMahon and Chapelle, 1991; Krumholz et al., 1997).

Previous laboratory heating experiments simulating the burial of marine sediments have shown an increase in acetate, which may potentially sustain the deep subseafloor biosphere (Wellsbury et al., 1997). Collectively, these data suggest that with



increasing depth, an increased organic carbon content provides microorganisms with more energy and carbon sources, and that the increased temperature accelerates the biodegradation of sedimentary organic matter and methanogenesis; as a result, the deepest aquifers in the Yuchi Formation function as sources of microbial methane. Acetate is considered a key intermediate
product, but as described above, acetoclastic methanogens constitute a minor proportion of the methanogens in the Yuchi Formation, and in our experiments under a high salinity condition, corresponding to that of the deepest sample, methanogenic activity from acetate decreased. We further demonstrate the potential for SAO coupled with hydrogenotrophic methanogenesis to convert acetate to methane in the deep part of this formation. Investigating the diversity of microorganisms involved in SAO in deep subsurface environments is among the targets for future study.

Our findings offer insight into microbial processes in the global carbon cycle over geological timescales and provide important reference data for geomicrobiological studies of deep subsurface environments that are enriched in microbial methane.

*Data Availability.* DNA sequencing data are available at GenBank, as described in the Material and Methods section. Other
datasets generated during the current study are available from the corresponding author on reasonable request.

*Competing Interests.* The authors have no relevant financial or non-financial interests to disclose.

*Author Contributions.* All authors contributed to the study conception and design. Taiki Katayama, Reo Ikawa, and Masaru
Koshigai collected the samples. Reo Ikawa and Masaru Koshigai analyzed the water and sediment geochemistry. Susumu Sakata analyzed the gas geochemistry. Taiki Katayama performed the cultivation experiments and the DNA sequencing analysis. All authors read and approved the final manuscript.

*Acknowledgments.* This study was carried out as a part of R&D supporting program titled "Development of enhancing the
evaluation technology for fresh-salt water interface in the coastal region" (2012 FY) under the contract with Ministry of Economy, Trade and Industry (METI). This study was also financially supported in part by Japan Society for the Promotion of Science KAKENHI grant numbers JP17K15183 and JP18H05295. We thank Hanako Mochimaru, Chiwaka Miyako, and Fumie Nozawa for assistance in the sample collection, sequencing analysis, and cultivation experiments. Thanks are further extended to Atsunao Marui for managing the borehole drilling project and to Yoichi Kamagata for valuable comments that
improved our manuscript.



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





**Table 1.** Geochemical characteristics of the water samples.

| Depth (mbgs) | pH | ORP (mV) | Temp (°C) | $HCO_3^-$ (mg L$^{-1}$) | $Cl^-$ (mg L$^{-1}$) | $NO_3^-$ (mg L$^{-1}$) | $Br^-$ (mg L$^{-1}$) | $SO_4^{2-}$ (mg L$^{-1}$) | $PO_4^{3-}$ (mg L$^{-1}$) |
|---|---|---|---|---|---|---|---|---|---|
| 100 | 7.2 | -200 | 10.6 | 453 | 165 | 1.4 | 1.4 | 0.6 | bdl |
| 149 | 7.3 | -210 | 12.1 | 395 | 306 | bdl | 2.8 | 0.1 | bdl |
| 476 | 8 | -490 | 22.2 | 637 | 1750 | bdl | 17 | bdl | 1.5 |
| 613 | 7.6 | -290 | 25.6 | 2150 | 9500 | bdl | 96 | bdl | 6.4 |
| 715 | 7 | -380 | 28.3 | 2980 | 15600 | bdl | 170 | bdl | 7.5 |
| 943 | 8.1 | -450 | 35.4 | 3610 | 17100 | bdl | 140 | bdl | bdl |

| Depth (mbgs) | $Na^+$ (mg L$^{-1}$) | $NH_4^+$ (mg L$^{-1}$) | $K^+$ (mg L$^{-1}$) | $Mg^{2+}$ (mg L$^{-1}$) | $Ca^{2+}$ (mg L$^{-1}$) | $Fe^{2+}$ (mg L$^{-1}$) | DOC (mg L$^{-1}$) | Ace. (mg L$^{-1}$) | δD (‰) |
|---|---|---|---|---|---|---|---|---|---|
| 100 | 146 | 4.3 | 22 | 38 | 37 | ndt | 7.5 | 0.2 | -79 |
| 149 | 160 | 15 | 20 | 66 | 50 | ndt | 5.2 | 0.029 | -76 |
| 476 | 1030 | 14 | 58 | 84 | 54 | 0.6 | 36 | 2.2 | -68 |
| 613 | 5640 | 79 | 230 | 270 | 89 | 1.4 | 85 | 0.63 | -46 |
| 715 | 9830 | 110 | 390 | 420 | 79 | ndt | 170 | 0.12 | -20 |
| 943 | 11100 | 210 | 440 | 310 | 100 | 1.6 | 220 | 16 | -12 |

Abbreviations: Ace., Acetate; bdl, below detection limit; DOC, Dissolved organic carbon; ORP, Oxidation-reduction potential; ndt, not determined; Temp, Temperature.





**Table 2.** Geochemical characteristics of the dissolved gas samples.

| Depth (mbgs) | Dissolved gas composition (%) | | | | Isotopic ratios (‰) | | |
| | | | | | CH$_4$ | | CO$_2$ |
| | N$_2$ | CO$_2$ | CH$_4$ | C$_2$H$_6$ | δD | δ$^{13}$C | δ$^{13}$C |
| --- | --- | --- | --- | --- | --- | --- | --- |
| 100 | 21.4 | 1.75 | 76.86 | 0 | −255 | −77.1 | −10.3 |
| 149 | 16.65 | 2.76 | 80.58 | 0 | −258 | −73.9 | −6.7 |
| 476 | 8.82 | 0.82 | 90.35 | 0 | −203 | −69.5 | −12.5 |
| 613 | 4.23 | 4.11 | 91.67 | 0 | −201 | −69.5 | −7.3 |
| 715 | 0.72 | 6.71 | 92.55 | 0.02 | −196 | −68.5 | −6.0 |
| 943 | 1.75 | 9.84 | 88.37 | 0.03 | −198 | −67.5 | −7.4 |




**Table 3.** Methanogen diversity based on the *mcrA* gene in the original water and culture samples.

| Representative clone ID in OTU | Accession no. | Related species | Identity (%) | Sarabetsu 100 | Sarabetsu 149 | Yuchi 476 | Yuchi 613 | Yuchi 715 | Yuchi 943 | H2/CO2 613 | H2/CO2 943 | Acetate 613 | Acetate 943 |
|---|---|---|---|---|---|---|---|---|---|---|---|---|---|
| | | | Depth (mbgs) | | | | | | | | | | |
| | | | Proportion (%) | Original water samples | | | | | | Culture samples Proportion (%) | | | |
| D3mf21 | LC214931 | *Methanosarcina mazei* | 97.8 | 2.9 | 2.9 | | | | | | | | |
| D14mf19 | LC214911 | *Methanosarcina subterranea* | 98.6 | | | 4.5 | 12.7 | | | | | | |
| D16Amf09 | LC214912 | *Methanosarcina subterranea* | 99.3 | | | | | | | 6.1 | | 90 | 86.8 |
| D13mf35 | LC214921 | *Methanolobus psychrophilus* | 99.3 | | | 7.5 | 3.2 | | 1.4 | | | | |
| D2mf17 | LC214928 | *Candidatus* Methanoperedens nitroreducens | 83.5 | 71 | 18.8 | | | | | | | | |
| D3mf15 | LC214933 | *Methanosaeta harundinacea* | 95.5 | 11.6 | 20.3 | | | | | | | | |
| D15mf27 | LC214924 | *Methanoregula formicica* | 84.9 | | | | 1.6 | 95.2 | | | | | |
| D3mf09 | LC214929 | *Methanoregula boonei* | 86.4 | 14.5 | 46.4 | | | | | | | | |
| D14mf27 | LC214922 | *Methanolinea mesophila* | 92.9 | | | 14.9 | 1.6 | | | | | | |
| D3mf29 | LC214934 | *Methanospirillum psychrodurum* | 100 | | 4.3 | | | | | | | | |
| D3mf07 | LC214932 | *Methanocalculus alkaliphilus* | 95.1 | | 4.3 | | | | | | | | |
| D16mf37 | LC214927 | *Methanoculleus sediminis* | 100 | | | | 11.1 | | 9.9 | | | | |
| D14Hmf08 | LC214914 | *Methanoculleus sediminis* | 100 | | | | | | | 36.4 | 27 | 5 | 2.6 |
| D16Hmf21 | LC214917 | *Methanoculleus bourgensis* | 92.9 | | | | | | | | 13.5 | | |
| D14Hmf25 | LC214915 | *Methanoculleus horonobensis* | 100 | | | | | | | 48.5 | 43.2 | | 5.3 |
| D16mf23 | LC214925 | *Methanoculleus horonobensis* | 99.3 | | | | 4.8 | | 12.7 | | | | |
| D3mf17 | LC214930 | *Methanobacterium alkalithermotolerans* | 100 | | 2.9 | | | | | | | | |
| D16mf27 | LC214926 | *Methanobacterium alkalithermotolerans* | 100 | | | 73.1 | 65.1 | 4.8 | 76.1 | | | | |
| D3Amf10 | LC214935 | *Methanobacterium alkalithermotolerans* | 100 | | | | | | | 9.1 | 16.2 | 5 | 5.3 |