# Peer review of "Microbial methane formation in deep aquifers associated with the sediment burial history at a coastal site"

_EGUsphere, 2023_

## Author Response (AR2)

**Responses to Referee #1**

We thank Referee #1 very much for his or her comments.

*Comments to the editor:*

The relevance oriented to natural gas exploitation is causing a moral conflict since a lot of emissions can be released by exploiting methane-rich sediments from aquifers. I believe the authors ought to adjust the relevance and use of the study to a more ecological perspective of the carbon cycle rather than more human exploitation of resources. After all the final highlight in the abstract is statements and mechanisms that are easily inferable without doing the research e.g., formation at highest temperature and migration through diffusion. The authors did not highlight anything relevant to gas exploitation in the end, so I deem it unnecessary at the start.

Response: In line with this comment, we removed the mention of the relevance to natural gas exploitation. For further details, please refer to our response to "*Comment to the authors 10*".

The manuscript is well written but there are certain nuisances in the style that require maybe a native English revision. Response: In accordance with this comment, we subjected the revised manuscript to English proofreading by two native speakers.

The authors purged the borewell in different ways and volume multipliers, which to me mean their samples run the risk of not
being purged for noise, good enough compromising their results. The authors need to explain this in detail to ensure their samples do bring a real view of the microbial geochemical cycling at the desired depths.

Response: Upon water sample collection, we verified the stabilization of specific chemical parameters. Additionally, the volume of water purged in advance aligns with the recommended criteria for water sample collection based on a previous study, assuring the representativeness of the samples. For further details, please refer to our response to "*Comment to the authors 75*".

I think authors need to double check the soundness of their analysis in terms of methanogen diversity, since there are new emerging criteria described by Prakash et al., Int. J. Syst. Evol. Microbiol. 2023;73:005500 DOI 10.1099/ijsem.0.005500. If authors document the precautions on their conclusions and analysis well, then this can be considered as valuable output, if not then major revision and new sequencing of the marker genes using more up-to-date primers is needed, or even metagenomics. The research is very centered in canonical archaea and methanogens within one phylum and the field is too advanced now to obviate or exclude a broader discussion, which is likely due to the limitation in the genetic markers they chose dating 2 decades. The discussion is too poor for what I expected, I gave points to discuss further and aspects from the methods to develop more and explain those gaps that for archaeal research and carbon cycling are lacking.

- 35 Response: The criteria described by Prakash et al. outline the minimal standards for the "taxonomic" description and classification of methanogens, which is beyond the scope of this study. Based on 16S rRNA gene sequencing, we checked the occurrence of microbes other than canonical archaea and methanogens within Euryarchaeota. Additionally, we cited and included information on newly discovered methane formation processes in recent studies. For further details, please refer to our responses to "*Comments to the authors 31-32, 115, and 230*".
- 40

**Comments to the authors:**

10: I invite the authors to remove "and to explore natural gas deposits", since this has a connotation of resource exploitation in a sector that already is a big emitter of GHG. The research should not highlight the use of methanogenesis ecological research as a survey tool for oil & gas industries, in particular when exploration will result in aquifer destruction.

Response: As suggested by the referee, we have removed the phrase "and to explore natural gas deposits".

18-19: I suggest to change "Cultivation of the hydrogenotrophic methanogens that dominated in the...." To "Cultivation of the dominant hydrogenotrophic methanogens in the..." it makes better sense.Response: As suggested by the referee, we have modified the sentence accordingly.

25-26: It would be better to leave it as the reservoir of organic matter where microbial processes drive geochemical cycling. The reference to a large fraction of the microorganisms present on earth is out of context or quantifiable reality. Response: As suggested by the referee, we have modified the sentence as follows:

Line 24-. Terrestrial subsurface environments are massive reservoirs of water and organic matter where microbial processes drive geochemical cycling (Lovely and Chapelle, Reviews of Geophysics, 33, 365-381, 1995; McMahon and Parnell, 2014). Aquifers that form in...

31-32: The authors have to take care of story development, since one sentence states that only CO2 remains as electron acceptor, yet later introduce the 3 methanogenic pathways where acetate of methylated compounds are also acceptors. I may add, all those are mcrA dependent and authors did not make any reference to novel methanogenic pathways recently discovered. They may not be archaea, or methanogenic for the purpose of energy conservation but they exist. Else, the authors should start the paragraph stating this is exclusively about archaea methanogenesis.

Response: As suggested by the referee, we have modified the relevant sentences as follows:

Line 29-. Methanogenesis, the process of methane formation by methanogens, represents a terminal step in the degradation of
 organic matter in anoxic environments (Lovely and Chapelle, Reviews of Geophysics, 33, 365-381, 1995). Methanogens form
 a diverse group of archaea that produce methane from various substrates, including H2 and CO2 (hydrogenotrophic),
 methylated compounds (methylotrophic), acetate (acetoclastic), methoxylated aromatic compounds (Mayumi et al., Science, 354, 222-225, 2016) or alkanes (Zhou et al., Nature, 601, 257-262, 2021).

Line 34-. In addition to methanogenesis, recent research has shed light on the process of methane formation by certain nonmethanogenic microbes, including cyanobacteria, algae, fungi, purple nonsulfur bacteria, and cryptogamic covers, which occur in oxygen-saturated aquatic and terrestrial ecosystems (Liu et al., Science of the Total Environment, 806, 151362, 2022).

35: loose sentence that does not integrate the story forming, it breaks narrative with the gas resources which to me is an energy-sector relevant sentence that should not mix with the science.

Response: As suggested by the referee, we have removed the sentence accordingly.

54:55: the depths 100, 149, 476, 613, 715, 943, should have an indication for resolution of the soil/rock type which visually is hard to discern from the figure.

Response: High-resolution lithology of the study sediments was previously demonstrated (Ikawa et al., 2014). In this manuscript, we have specifically described the lithology of sampled aquifers as follows:

- Line 56. In this study, water samples were collected from the saline aquifers in the Yuchi Formation [at 476 (fine sand), 613 (coarse sand), 715 (fine sand), 943 (fine sand) mbgs] and from the freshwater aquifers in the upper part of the Sarabetsu Formation [at 100 (coarse sand), and 149 (coarse sand) mbgs], as reference...
- 85 75: The volumes of the wellbore were extracted in different excess, why? This means you did not use the same criteria for extraction of "noise" water volume. Is 4 times sufficient to purge the water that is not representing the desired depth? In addition, how was this applied to all different depths? This is crucial for the soundness of your study. Response: In this study, the criterion for water sample collection is based on the stabilization of specific chemical parameters, including the water temperature, electrical conductivity, pH, and oxidation–reduction potential, as described in Lines 78-79.
- 90 To ensure that the sampled groundwater is indeed from the aquifer itself, it is standard practice to pump 3-5 times the volume of water in the wellbore or the area sealed off by packers, as recommended by Kieft et al. in "*Manual of Environmental Microbiology*" (pp. 799-817, ASM Press, 2007). The water volume pumped from the well in our study also meets this criterion.

**79: filled with water to maintain anaerobic conditions? But the samples are.... Water? This is very confusing.**

Response: The bottle was filled with a "water sample" to raise the internal pressure of the bottle and prevent the penetration of air (oxygen). We have modified the relevant sentences as follows:
 Line 82-. Water samples for microbial cultures were collected in sterilized glass bottles with butyl rubber stoppers and screw caps. The bottles were purged with N2 gas before and during sample collection and then filled with the water sample to raise the internal pressure of the bottles, effectively preventing the penetration of air.

*83-84: the gas separation during sampling should be explained in detail, not all readers are experts in these methods.* Response: As suggested by the referee, we have modified the relevant sentences as follows: Line 87-. The gases that were associated with the formation water in aquifers (i.e., under high-pressure conditions) were naturally separated from the water when it was sampled under atmospheric pressure. These separated gases were collected using the water displacement method.

**90-92: details on the column could be useful to include.**

Response: The description of the gas composition and isotope ratio measurement methods has been revised to include information about the analysis columns, as follows:

Line 95-. The gas composition was analyzed using two gas chromatographs: a Shimadzu GC-8A equipped with a thermal conductivity detector and a Molecular Sieve 5A column and an Agilent 6890 equipped with a flame ionization detector and a PoraPLOT Q column. Stable carbon ( $\delta^{13}$ C) and hydrogen ( $\delta$ D) isotope ratios of methane, along with the  $\delta^{13}$ C of carbon dioxide, were measured using a Trace Ultra gas chromatograph connected to a DELTA V plus isotope ratio mass spectrometer (IRMS) through a GC IsoLink combustion/pyrolysis interface (Thermo Fisher Scientific), and the column was a PoraPLOT Q.

**115**

115: perhaps a more expert opinion is needed, but I do NOT think the primer selection and references for microbial analysis are up to date. What about the former Rice ClusterI? Now to be Methanomassiliicocci? Also the use of mcrA gene markers as a stand-alone method to classify has been flagged as non sufficient by the recent Proposed minimal standards for description of methanogenic archaea. I suggest the authors review this recommendation and describe the fulfilment with discussion around their findings according to emerging needs to harmonize the way methanogens are described and reported: Prakash et al., Int. J. Syst. Evol. Microbiol. 2023;73:005500 DOI 10.1099/ijsem.0.005500

Response: We thank the referee for providing this valuable feedback. First, we wanted to confirm that our analysis encompasses not only mcrA gene sequences but also 16S rRNA gene sequences. For a comprehensive understanding, we direct the referee to the dedicated section "3.3 Microbial Community Compositions in the Water Samples", where detailed information is available.

Second, we appreciate his or her reference to the paper by Prakash et al. outlining the minimal standards for "taxonomic" description and classification of methanogens. However, it is important to clarify that our study is centered around the population, activity, and metabolic pathway of methanogens in aquifers with varying temperature and salinity concentrations. Taxonomic description and classification of methanogens are beyond the scope of this study.

Third, we acknowledge the recent reclassification of the former Rice Cluster I as Methanocella. Upon reviewing our data, we confirmed the absence of Methanocella in our samples.

**154: not methylated compounds? Like methylamines?**

Response: In this study, we did not attempt to culture methylotrophic methanogens using methylated compounds as
methanogenic substrates, as our primary focus was not on this pathway of methanogenesis. As supported by the 16S rRNA gene sequencing analysis, methylotrophic methanogens were found to be present in only minor proportions (Fig. 3c).

230: The authors should make a discussion around other methanogens in reference to their gene primers dating almost 2 decades ago and also on new markers used in more recent research. What about Bathyarchaeota? All your findings come from Euryarchaeota and this seems biased to your methods. What about the Asgard superphylum? To give a more holistic view of carbon cycling in these environments? This can be regardless of methanogenic potential but a complete view of the diversity likely missed by your gene markers of choice.

Response: As previously mentioned, we analyzed 16S rRNA gene amplicon sequences targeting both bacteria and archaea for the water samples (Fig. 3). Consequently, certain potential methanogens possessing the Mcr gene but not classified within

- 145 Euryarchaeota, such as Methanomethylicales, Nezhaarchaeales, Korarchaeia and Methanodesulfokores, were not detected in 16S rRNA gene sequencing reads. Additionally, the 16S rRNA gene sequences that were closely related to those of Bathyarchaeota and Asgard, known to possess Acr but not Mcr (for more information, refer to Mei et al., PNAS Nexus, 2, pgad023, 2023), were not detected in this study either. We therefore conducted traditional mcrA gene sequencing to investigate whether the diversity of methanogens observed in the 16S rRNA gene analysis was also present in the mcrA gene analysis. To
- 150 incorporate the above information into the manuscript, we have created the following paragraph: Line 252-. Apart from the conventional methanogens mentioned above, recent studies indicate the potential of archaea outside the classification of Euryarchaeota to produce methane or breakdown nonmethane alkanes. Specifically, *Methanomethylicales*, *Nezhaarchaeales*, *Korarchaeia*, and *Methanodesulfokores* have been found to possess methyl-S-CoM reductase, the central enzyme of methanogenesis, while *Bathyarchaeota* and *Asgard* possess alkyl-S-CoM reductase (Acr), which activates
- 155 nonmethane alkanes (Mei et al., PNAS Nexus, 2, pgad023, 2023). In the archaeal 16S RNA gene sequences of our samples, however, none of these potential methanogens or hydrocarbon-oxidizing archaea were identified.
   With regard to the topic of carbon cycling, our investigation detected *Ca. Methanoperedens*, a microorganism capable of methane oxidation through coupling with nitrate reduction. We explored the importance of this process by examining the relationship between δ13C-CH4 and δ13C-CO2 (refer to Lines 264-266).

**Responses to Referee #2**

**GENERAL COMMENTS**

The manuscript by Katayama et al., entitled "Microbial methane formation in deep aquifers associated with the sediment burial history at a coastal site," presents an interesting study that analyzes the origin of methane in a natural gas-bearing

- 165 sedimentary aquifer, located in a coastal area of northern Japan. The authors have used an elegant approach: They have investigated a special geological setting in which two aquifers of different geochemical composition are efficiently isolated by a clay aquitard. They have built their research on the careful characterization of the site in a previous study (Ikawa et al., 2014), and conducted a new sampling campaign to collect high-quality, large-volume water samples from six carefully selected target depths, up to 943 m deep below ground surface and with in situ temperatures up to 35°C. They have investigated these
- 170 samples with state-of-the art methods, using stable isotope analyses of methane and water, DNA-based analysis of microbial communities in the pristine water samples, and additional cultivation experiments.

The outstanding strength of this study is the selection of an ideal study site in a geological setting, which exhibits a distinct depth profile for the hydrogen isotopic composition of water. Thanks to this fact, the authors cannot only identify microbial CO2 reduction as the predominant pathway of methanogenesis, but also delineate the depth of the methane source. They show convincingly that methanogenesis is highest in the deepest aquifer, where water temperature, salinity, and total organic carbon contents of the adjacent mud sediments were highest.

The scientific questions are relevant and within the scope of BG. The concept and data are novel, and the conclusion is substantial. The results are discussed in an appropriate and balanced way. Overall, the manuscript is well organized, reads well and I couldn't detect major scientific flaws. The topic is of interest to scientists both in the CH4 community and in the field of deep biosphere research. My recommendation is: Essentially, the paper can be published with some minor revisions as suggested below.

Response: We thank the referee very much for his or her thoughtful and positive evaluation of our manuscript. The careful characterization of the study site and the utilization of stable isotope analyses, DNA-based microbial community analysis, and cultivation experiments were integral to our research, and we are glad to see these aspects recognized.

- 185 We also greatly appreciate the comments highlighting the distinct depth profile for the hydrogen isotopic composition (methane vs. water) as a key factor contributing to the strength of our study. It is rewarding to know that this factor enabled us to not only identify the dominant pathway of methanogenesis but also pinpoint the depth of the methane source. Furthermore, we extend our sincere gratitude for the suggestion that the paper can be published with minor revisions. We have carefully incorporated the recommended revisions into the manuscript, as outlined below.
- 190

**MINOR COMMENTS**

*L.* 37/Fig. S1: Location of the investigated site. To improve readability, please add another map with a larger scale to show the location of the investigated site in the northern part of Hokkaido/within Japan.

Response: As suggested by the referee, we have added another map with a larger scale showing the entire Hokkaido region in 195 Fig. S1.

*L.* 47: "Throughout the burial history of the Yuchi Formation, water salinity decreased while temperature increased along the geothermal gradient (Fig. 1d)." – This sentence is misleading. I don't think that salinity decreased with the geothermal gradient. The opposite is true. Salinity increases with depth. Please clarify this statement. Two sentences might be better than one.

Response: Certainly, the statement in question appears to contain a potential misinterpretation. Considering the earlier descriptions of salinity gradient formation (Lines 40-48 in the revised manuscript), we have revised this sentence to solely focus on the change in temperature with depth, as follows:

Line 48-. Throughout the burial history of the Yuchi Formation, temperature increased along the geothermal gradient (Fig. 1d).

**L. 63/64: Please add information about the location of the site on Hokkaido, Japan.**

Response: As suggested by the referee, we have modified the relevant sentences as follows:

Line 66-. The study site was located on a sand dune 300 m from the coastline of the Teshio Plain located in northern Hokkaido 210 (44.9948° N, 141.6882° E) at an elevation of 5.2 m above sea level (Fig. S1).

*L.* 198: "In all water samples, CH4 accounted for approximately >75% of the total dissolved gas (Table 2)." This is the analytical result for the recovered gas phase. Please add information on the concentration of dissolved methane in the aqueous phase (in text and table). The conversion requires a mass balance calculation (total amount of released methane (moles)

divided by water volume from which the gas was released).

Response: The concentration of dissolved methane in the aqueous phase serves as an indicator reflecting the history of biogenic methane formation within the sediment. We agree with the perspective of Referee 2, acknowledging that incorporating such information could have strengthened the validity of the discussion presented in this paper. Regrettably, due to several constraints during field operations and the limited anticipation of its significance at the time of sample collection, measurements of dissolved methane concentration were not conducted. Consequently, we are unable to provide additional data on the concentration of dissolved methane in the acueous phase. We concentrate Deforme 21a incidentful characteristic and on the concentration of dissolved methane in the aqueous phase. We appreciate Referee 2's insightful observation and understanding of the circumstances that hinder our ability to address this point.

L. 204-207: "Methane dissolved in water from the Koetoi Formation, which underlies the Yuchi Formation (Fig. S1), plotted
 near the boundary between a biogenic and a thermogenic origin (Tamamura et al., 2014) (Fig. 2a). The lack of thermogenic methane produced at great depth in the upper Sarabetsu and Yuchi Formation samples implies that methanogenesis occurred within these formations." – This part could be refined. I understand that methane in the Koetoi Fromation has a distinctly different isotopic composition. The isotopic difference implies, that methane in the Yuchi Formation is not derived from thermogenic sources in the Koetoi formation. Please include some information on the line of arguments, if it is correct. L. 209-

214/Fig. 2: Please use symbols instead of the words "squares", "circles" and "triangels" to increase (quick) readability.Response: As suggested by the referee, we have modified the relevant sentences as follows:

Line 210-. Methane dissolved in water from the Koetoi Formation, which underlies the Yuchi Formation (Fig. S1), plotted near the boundary between a biogenic and a thermogenic origin (Tamamura et al., 2014) (Fig. 2a). The isotopic difference indicates that the proportion of methane in the upper Sarabetsu and Yuchi Formations derived from the lower Koetoi Formation was minimal.

In accordance with the suggestion, we have now substituted the terms "squares," "circles," and "triangles" with their corresponding symbols in the caption of Figure 2.

*L.* 306: "*A remarkably high methanogen population* [...]" – *Why was the finding remarkable? What was the expectation?*

Response: Based on this comment, we recognize that the term "remarkably" might lead to confusion. Consequently, we have replaced it with "distinctly".

*L. 315:* "Previous laboratory heating experiments simulating the burial of marine sediments have shown an increase in acetate, which may potentially sustain the deep subseafloor biosphere (Wellsbury et al., 1997)." – For your information: The effect of heating on acetate concentrations, a substantial stimulation of microbial activity, and the role of SAO have also been shown in deeply buried marine sediments with the help of pore-water depth profiles and radiotracer experiments, e.g. Heuer et al. (2020) and Beulig et al., (2022).

• Heuer, V.B., Inagaki, F., Morono, Y., Kubo, Y., Spivack, A.J., Viehweger, B., Treude, T., Beulig, F., Schubotz, F., Tonai, S., Bowden, S.A., Cramm, M., Henkel, S., Hirose, T., Homola, K., Hoshino, T., Jiiri, A., Imachi, H., Kamiva, N., Kaneko, M.,

- Lagostina, L., Manners, H., McClelland, H.L., Metcalfe, K., Okutsu, N., Pan, D., Raudsepp, M.J., Sauvage, J., Tsang, M.Y., Wang, D.T., Whitaker, E., Yamamoto, Y., Yang, K.H., Maeda, L., Adhikari, R.R., Glombitza, C., Hamada, Y., Kallmeyer, J., Wendt, J., Wormer, L., Yamada, Y., Kinoshita, M. and Hinrichs, K.U. (2020) Temperature limits to deep subseafloor life in the Nankai Trough subduction zone. Science 370, 1230-1234. doi: 10.1126/science.abd7934
  Beulig, F., Schubert, F., Adhikari, R.R., Glombitza, C., Heuer, V.B., Hinrichs, K.U., Homola, K.L., Inagaki, F., Jorgensen,
- 255 B.B., Kallmeyer, J., Krause, S.J.E., Morono, Y., Sauvage, J., Spivack, A.J. and Treude, T. (2022) Rapid metabolism fosters microbial survival in the deep, hot subseafloor biosphere. Nature Communications 13. doi: 10.1038/s41467-021-27802-7 Response: We thank the referee for bringing up this valuable point. His or her references to the pore-water depth profiles and radiotracer experiments conducted by Heuer et al. (2020) and Beulig et al. (2022) indeed provide further insight into the influence of heating on acetate concentrations and the potential role of SAO on microbial activity in deeply buried sediments.
- 260 We have therefore added these references and revised the relevant sentences as follows:

Line 324-. Previous laboratory heating experiments simulating the burial of marine sediments have shown an increase in acetate, which may sustain the deep subseafloor biosphere (Wellsbury et al., 1997). Indeed, the porewater acetate concentration in deep subseafloor sediments increases with depth, which stimulates potential methanogenic activities (Heuer et al., 2020; Beulig et al., 2022).

Line 335-. We further demonstrate the SAO activity coupled with hydrogenotrophic methanogenesis to convert acetate to methane in the sample from the deep part of this formation. A previous study suggested the potential for SAO with methanogenesis in deep subseafloor sediments at high temperatures (Beulig et al., 2022). Investigating the diversity...

*L.* 534/Table 2: For the data on isotopic compositions, please include information on the reference standards against which the delta-values are reported.

Response: As suggested by the referee, we have added information on the reference standards (VPDB and VSMOW for carbon and hydrogen isotopic compositions, respectively) in Tables 1 and 2.